# Features of patients that died for COVID-19 in a hospital in the south of Mexico: A observational cohort study

**Jesús Arturo Ruíz-Quiñonez**[1☯], **Crystell Guadalupe Guzmán-Priego**[2☯], **Germán Alberto Nolasco-Rosales**[2], **Carlos Alfonso Tovilla-Zarate**[3], **Oscar Israel Flores-Barrientos**[1], **Víctor Narváez-Osorio**[1], **Guadalupe del Carmen Baeza-Flores**[2], **Thelma Beatriz Gonzalez-Castro**[4], **Carlos Ramón López-Brito**[1], **Carlos Alberto Denis-García**[1], **Agustín Pérez-García**[1], **Isela Esther Juárez-Rojop**[2]*

1 Secretaría de Salud, Hospital de Alta Especialidad Dr. Juan Graham Casasús, Villahermosa, Tabasco, México, 2 División Académica de Ciencias de la Salud, Universidad Juárez Autónoma de Tabasco, Villahermosa, Tabasco, México, 3 División Académica Multidisciplinaria de Comalcalco, Universidad Juárez Autónoma de Tabasco, Comalcalco, Tabasco, México, 4 División Académica de Multidisciplinaria de Jalpa de Méndez, Universidad Juárez Autónoma de Tabasco, Jalpa de Méndez, Tabasco, México

☯ These authors contributed equally to this work.
* iselajuarezrojop@hotmail.com

## Abstract

### Background

Due to the wide spread of SARS-CoV2 around the world, the risk of death in individuals with metabolic comorbidities has dangerously increased. Mexico has a high number of infected individuals and deaths by COVID-19 as well as an important burden of metabolic diseases; nevertheless, reports about features of Mexican individuals with COVID-19 are scarce. The aim of this study was to evaluate demographic features, clinical characteristics and the pharmacological treatment of individuals who died by COVID-19 in the south of Mexico.

### Methods

We performed an observational study including the information of 185 deceased individuals with confirmed diagnoses of COVID-19. Data were retrieved from medical records. Categorical data were expressed as proportions (%) and numerical data were expressed as mean ± standard deviation. Comorbidities and overlapping symptoms were plotted as Venn diagrams. Drug clusters were plotted as dendrograms.

### Results

The mean age was 59.53 years. There was a male predominance (60.1%). The mean hospital stay was 4.75 ± 4.43 days. The most frequent symptoms were dyspnea (88.77%), fever (71.42%) and dry cough (64.28%). Present comorbidities included diabetes (60.63%), hypertension (59.57%) and obesity (43.61%). The main drugs used for treating COVID-19 were azithromycin (60.6%), hydroxychloroquine (53.0%) and oseltamivir (27.3%).

**Data Availability Statement:** All data are available at Mendeley Data (http://dx.doi.org/10.17632/cyky2m38g5.1).

**Funding:** The authors received no specific funding for this work.

## Conclusions

Mexican individuals who died of COVID-19 had shorter hospital stays, higher frequency of shortness of breath, and higher prevalence of diabetes than individuals from other countries. Also, there was a high frequency of off-label use of drugs for their treatment.

## Introduction

On December 31, 2019, the World Health Organization (WHO) was informed of cases of pneumonia of unknown etiology in Wuhan, China. The Chinese authorities identified a new type of coronavirus, which was isolated on January 7, 2020 [1]. Coronaviruses are enveloped RNA viruses, which are the cause of SARS (Severe Acute Respiratory Syndrome) and MERS (Middle East Respiratory Syndrome) in humans. The novel coronavirus was named SARS-CoV2 and COVID-19 is the name of the disease [2, 3]. Since it was discovered, SARS-CoV-2 has widely spread around the world; by August 15[th] 2020, there were 21,037,564 confirmed cases and 755,455 deaths reported worldwide [4].

To date, there are some reports and reviews about the characteristics of those who have had COVID-19 in a few populations around the world. These reports agree that the median age in individuals who were hospitalized due to COVID-19 was between 47 and 73 years, with a male predominance (60%). Also, the most common symptoms were fever (90%), dry cough (60–86%), shortness of breath (53–80%), fatigue (38%), nausea/vomiting or diarrhea (15–39%), and myalgia (15–44%). The most frequent comorbidities observed included hypertension (48–57%), diabetes (17–34%) and cardiovascular diseases (21–28%). They also reported that comorbidities were more common in hospitalized individuals (60–90%) compared with the overall population infected by COVID-19 (25%) [5].

In Mexico, the first case of COVID-19 was detected on February 27, 2020 [6]; by August 15[th], there were 505,751 confirmed cases and 55,293 deaths [4]. Because there is an important burden of metabolic diseases such as diabetes (10.3%), hypertension (18.4%) and obesity (36.1%) in the Mexican population [7], this population could be highly vulnerable to COVID-19.

Up to today, there are very few reports in the Mexican population that have used epidemiological data to evaluate comorbidities as risk factors of COVID-19 [8, 9]. For instance, we found only one report that evaluated gastrointestinal symptoms in Mexican individuals with COVID-19 [10].

We consider that is also important to evaluate the pharmacological treatment most commonly used in Mexico to treat COVID-19. Although there is not directed antiviral therapy recommended against SARS-CoV-2 [11–13], it is common knowledge that Mexican hospital include the use of antivirals for the treatment of individuals with COVID-19. Therefore, our aim was to evaluate demographic features, clinical characteristics, and pharmacological treatment received by individuals who died by COVID-19 in a third level hospital in the south of Mexico.

## Methods

This was an observational retrospective study performed at the High-Specialty Regional Hospital "Dr. Juan Graham Casasús" (HJGC) in Villahermosa, Tabasco, Mexico. We used the medical records of every individual who died by COVID-19 and was admitted to hospital

(HJGC) from April 15th to May 12th, 2020. These individuals were diagnosed with COVID-19 by epidemiologists following the Mexican Health Secretary's guidelines; additionally, they tested positive for SARS-CoV-2 by Reverse Transcriptase Polymerase Chain Reaction. The ethical approval was granted by the Research Ethics Committee of the Juarez Autonomous University of Tabasco (103/CIP-DACS/2020) in Mexico, and ethics committee of HJGC waived the requirement for informed consent.

Three physician doctors retrieved data from clinical files after patients' death. Data collected included age, gender, clinical symptoms, underlying comorbidities, length of hospital stay, and drugs used for the treatment of COVID-19.

We performed a descriptive report. Numerical data were expressed as mean ± standard deviation; categorical data were expressed as proportions (%). The distribution of gender, age and comorbidities were compared using the Chi-squared test, with significance of p = 0.05. Comorbidities and overlapping symptoms where plotted as Venn diagrams. Drug clusters were plotted as dendrograms. All data were analyzed using SPSS v.23.

## Results

We selected the files of 185 deceased individuals who had COVID-19; these individuals had been diagnosed and treated for COVID-19 between April 15 and May 12, 2020 at the HJGC in Villahermosa Tabasco, Mexico. The length of time hospitalized was 4.75 days in means (S.D. 4.43). The distribution of days of hospitalization by gender is shown in Fig 1. All patients included in this study died of cardiorespiratory arrest, secondary to acute respiratory failure.

### Age and gender

The mean age of the individuals studied was 59.53±12.50 (range 24–86) years. When we stratified them by age, the majority were individuals of 65+ years (36.7%, n = 58). Nevertheless, the groups of 55 and 65+ years constituted 67.7% of the sample (n = 107). The comparison between males and females is shown in Table 1. No significant differences were observed between groups in terms of age.

### Symptoms

The most common symptoms on admission were shortness of breath (88.77%), fever (71.42%), dry cough (64.28%), headache (43.87%) and myalgia (34.69%). Frequencies per gender are shown in Table 2. When overlapping symptoms were observed, the first cluster included fever, dry cough, shortness of breath and headache. Another cluster included fever, dry cough, shortness of breath, myalgia, arthralgia, and headache. Then fever, dry cough, shortness of breath, myalgia, and arthralgia without headache. And fever and shortness of breath (Fig 2).

### Comorbidities

The most common comorbidities on admission were Type 2 Diabetes (60.63%), hypertension (59.57%) and obesity (43.61%) (Table 1). The main overlap between these comorbidities were T2D-Hypertensión. The second overlap was T2DM-Hypertension-Obesity and finally Hypertension-Obesity (Fig 3). Other less frequent comorbidities observed in individuals who died by COVID-19 are depicted in Table 3. There was one individual with HIV and other two with cancer; the three of them were immunosuppressed. It is important to mention that these individuals were also receiving the usual treatment for their underlying disease while they were treated in hospital for COVID-19.

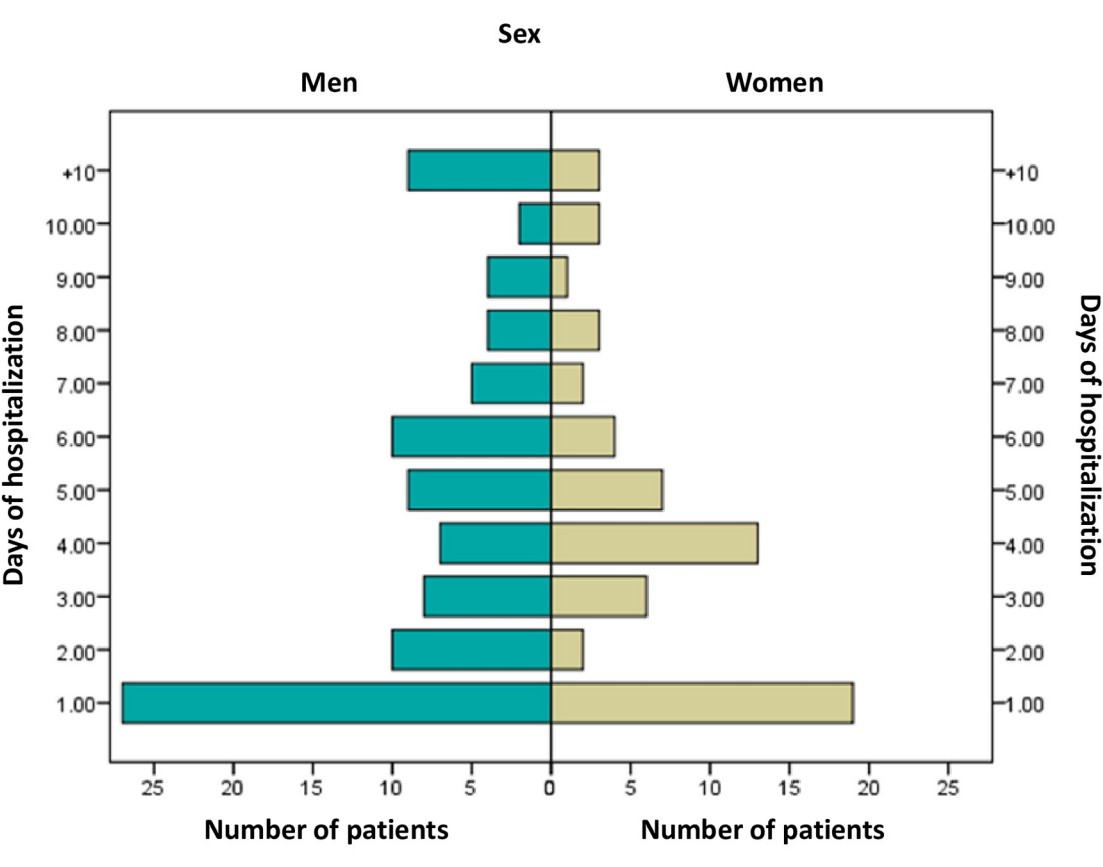

**Fig 1. Individuals who died of COVID-19 stratified by gender and days of hospitalization.**

**Table 1. Characteristics stratified by gender of individuals who died of COVID-19.**

|  | All | Male | Female | X², p |
|---|---|---|---|---|
| **Total** | 158 | 95 (60.1%) | 63 (39.9%) |  |
| **Age** |  |  |  | 4.36, 0.35 |
| 24–34 | 5 (3.2%) | 3 (3.2%) | 2 (3.2%) |  |
| 35–44 | 13 (8.2%) | 9 (9.5%) | 4 (6.3%) |  |
| 44–54 | 33 (20.9%) | 24 (25.3%) | 9 (14.3%) |  |
| 55–64 | 49 (31.0%) | 29 (30.5%) | 20 (31.7%) |  |
| 65+ | 58 (36.7%) | 30 (31.6%) | 28 (44.4%) |  |
| **Comorbidities** |  |  |  | 2.55, 0.11 |
| Any | 101 (63.9%) | 56 (58.9%) | 45 (71.4%) |  |
| No or not reported | 57 (36.1%) | 39 (41.1%) | 18 (28.6%) |  |
| Type 2 Diabetes | 56 (35.4%) | 27 (27.4%) | 29 (46.0%) | 1.83, 0.17 |
| Obesity | 41 (25.9%) | 21 (22.1%) | 20 (31.7%) | 5.13, 0.12 |
| Hypertension | 56 (35.4%) | 31 (32.6%) | 25 (39.7%) | 0.82, 0.36 |

**Table 2. Symptoms frequencies stratified by gender of individuals who died of COVID-19.**

|  | All (n = 98) | Male (%) | Female (%) |
|---|---|---|---|
| **Shortness of breath** | 87 (88.77%) | 48 (55.2%) | 39 (44.8%) |
| **Fever** | 70 (71.42%) | 36 (51.4%) | 34 (48.6%) |
| **Dry cough** | 63 (64.28%) | 37 (58.7%) | 26 (41.3%) |
| **Headache** | 43 (43.87%) | 24 (55.8%) | 19 (44.2%) |
| **Myalgia** | 34 (34.69%) | 15 (44.1%) | 19 (55.9%) |
| **Arthralgia** | 33 (33.67%) | 17 (51.52%) | 16 (48.48%) |
| **Rhinorrhea** | 16 (16.33%) | 8 (50.00%) | 8 (50.00%) |
| **Diarrhea** | 16 (16.33%) | 10 (62.50%) | 6 (37.50%) |
| **Sputum** | 7 (7.14%) | 5 (71.43%) | 2 (28.57%) |

## Pharmacological therapy

The most frequent drugs used for treating individuals with COVID-19 were azithromycin (69.32%) and hydroxychloroquine (61.36%). Less frequently employed were oseltamivir (38.64%), lopinavir-ritonavir (26.14%) and tocilizumab (22.73%). Drugs for various purposes (e.g. paracetamol, dexamethasone, enoxaparin) were grouped as others, these were given to 73.86% of the patients (Table 4).

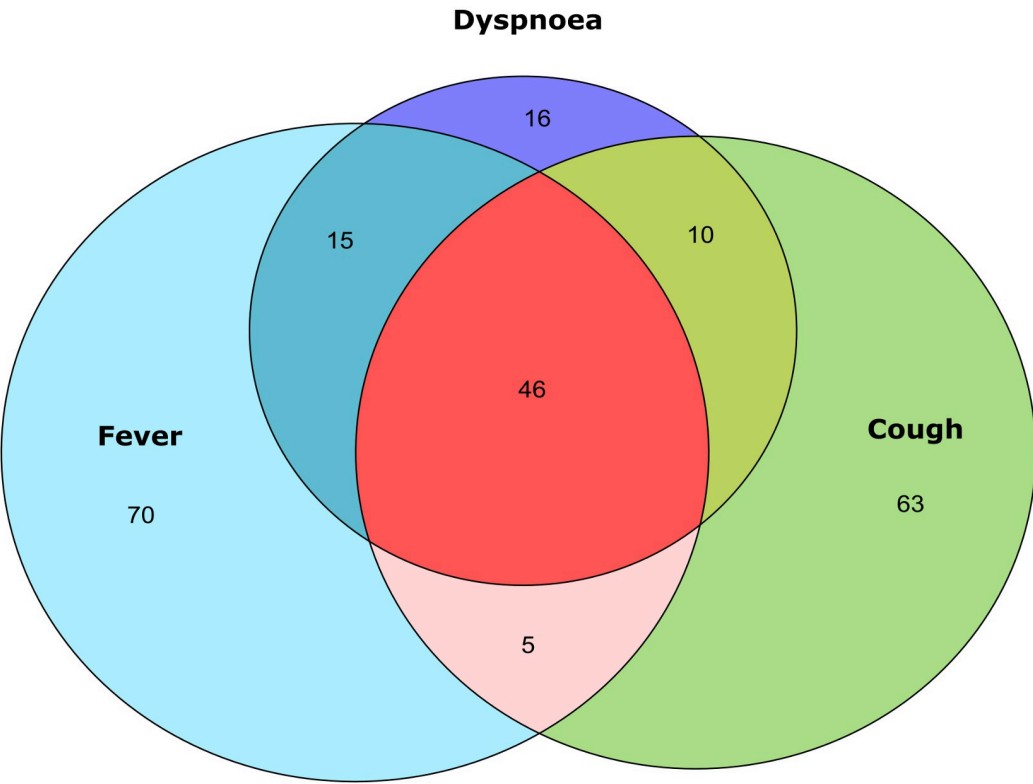

**Fig 2. Venn diagram of overlapping symptoms.** We plotted overlaps between the three main symptoms reported in the sample. The higher proportions were fever or cough only, without manifestation of another symptom. The largest overlap was the triad fever-cough-dyspnea. Then, in order of frequency the main overlaps were fever-dyspnea and cough-dyspnea.

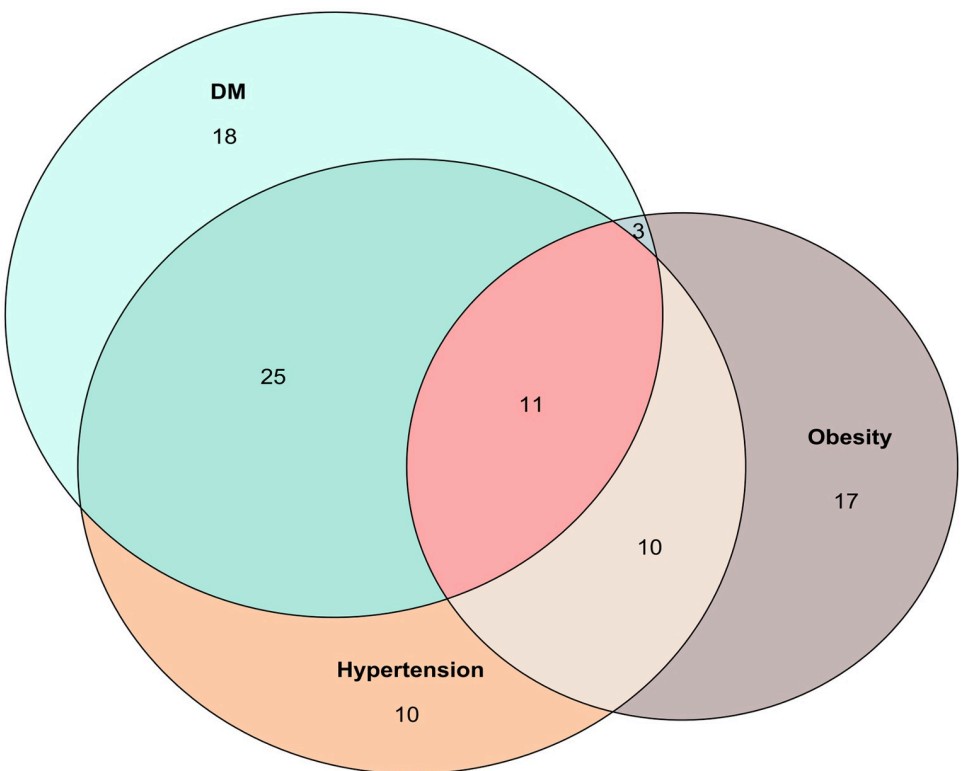

**Fig 3. Venn diagram of overlapping comorbidities.** The main overlap was the dyad Hypertension-T2D. The frequency of these comorbidities alone was almost equal. The next overlaps in order of frequency were the triad Hypertension-T2D-Obesity and the dyad Hypertension-Obesity.

We observed that azithromycin and hydroxychloroquine were frequently given together, followed by lopinavir-ritonavir with tocilizumab, and oseltamivir with lopinavir-ritonavir or tocilizumab (Fig 4).

## Discussion

In this study, we gathered clinical and demographic characteristics of individuals who died of COVID-19; we also identified drugs that were most frequently used as antiviral treatment in those individuals.

According to Mexico's epidemiological data [8], there is a male predominance for COVID-19. Then, having 60% males in our sample was expected. Furthermore, this predominance is

**Table 3. Other comorbidities of individuals who died of COVID-19.**

|  | All (n = 158) | Male | Female |
|---|---|---|---|
| **Coronary disease** | 3 | 1 | 2 |
| **Chronic kidney disease** | 9 | 4 | 5 |
| **Asthma** | 4 | 1 | 3 |
| **HIV** | 1 | 1 | 0 |
| **Cancer** | 2 | 0 | 2 |
| **Alcoholism** | 1 | 1 | 0 |
| **Tobacco use** | 5 | 5 | 0 |

**Table 4. Drugs most commonly used for the treatment of COVID-19.**

|  | All (n = 88) | Male (%) | Female (%) |
|---|---|---|---|
| **Azithromycin** | 61 (69.32%) | 32 (52.46%) | 29 (47.54%) |
| **Hydroxychloroquine** | 54 (61.36%) | 28 (51.85%) | 26 (48.15%) |
| **Oseltamivir** | 34 (38.64%) | 20 (58.82%) | 14 (41.18%) |
| **Lopinavir-ritonavir** | 23 (26.14%) | 10 (43.48%) | 13 (56.52%) |
| **Tocilizumab** | 20 (22.73%) | 14 (70.00%) | 6 (30.00%) |
| **Others** | 65 (73.86%) | 36 (55.38%) | 29 (44.62%) |

also observed in other populations including Chinese and British [5, 14–16]. We found that 67.7% of individuals were ≥55 years old. Similar findings had been observed in the state of Sonora, Northwest of Mexico, with a mean age of 56.98 years old in individuals who died by COVID-19 [17]. On the other hand, reports in the USA population indicated that the majority of individuals who have died of COVID-19 were also males (53.3%), and aged ≥65 years (78.2%) [18]. Although age varied widely among reports, we consider that our results still reflect similar demographic characteristics to patients with severe COVID-19 in other populations [14, 19]. The mean length time hospitalized in days was $4.75 \pm 4.43$, which is shorter than what was reported in an observational study of fatal cases from Wuhan, China ($6.35 \pm 4.51$) [19].

Shortness of breath, fever and dry cough were the most prevalent symptoms in our sample. Fever and cough had been reported as the most prevalent symptoms in case series from Europe, China, Mexico and in severe cases [5, 10, 15, 19–21]. On the other hand, the third most prevalent symptom varied widely; for instance, a review found dyspnea in 53–80% of cases, while a series of fatal cases in Wuhan found dyspnea in 70.6% [5, 19]. Nonetheless, in clinical series and meta-analysis involving non-lethal cases in Asia, Europe, and North

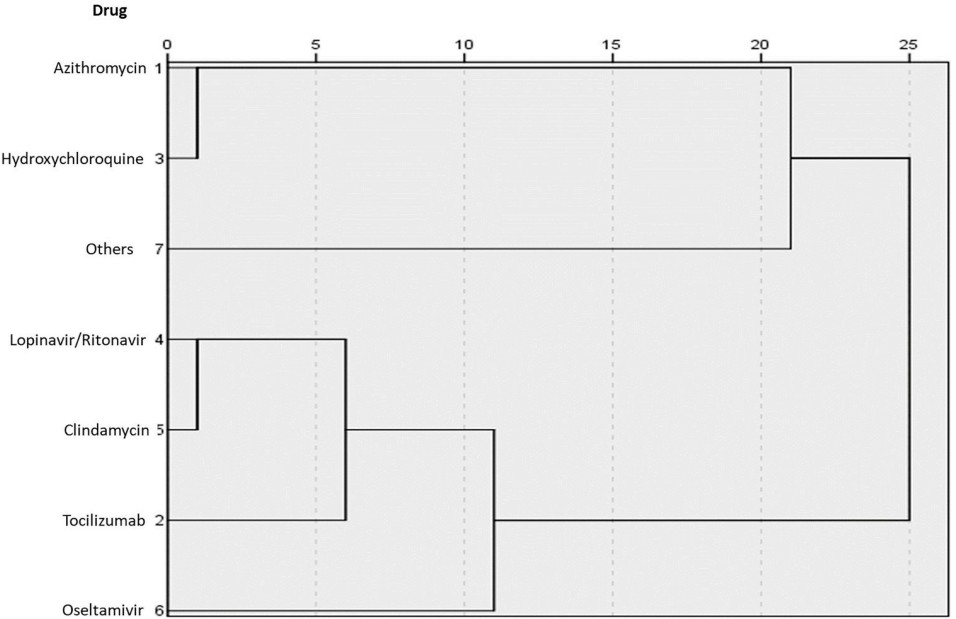

**Fig 4. Dendrogram of relationships between drugs used.** The closest relation was azithromycin–hydroxychloroquine, followed by oseltamivir–lopinavir or tocilizumab.

America, the third most prevalent symptom was myalgias or fatigue (31–67%) [10, 15, 20, 21]. In our sample, shortness of breath was the most prevalent symptom, and it was present in every cluster of symptoms, which could indicate that shortness of breath is an early symptom of bad prognosis.

The most common comorbidities in our sample were type 2 diabetes, hypertension, and obesity. Remarkably, diabetes and hypertension were found in similar frequencies. It has been reported that in Mexican individuals with COVID-19 there is a slightly higher frequency of hypertension than diabetes [8, 10]. Additionally, a report from Northwest Mexico indicated that diabetes and hypertension have higher risk of mortality than other comorbidities [17]. In a follow-up study of 20,133 individuals with COVID-19 in United Kingdom, the most frequent comorbidities were chronic cardiac disease (31%), uncomplicated diabetes (21%), non-asthmatic chronic pulmonary disease (18%), and chronic kidney disease (16%); while only 23% reported no major comorbidities [22]. On the other hand, a report in Northeast Brazil showed that the mortality of COVID-19 was enhanced by old age, neurological diseases, pneumopathies and cardiovascular diseases [23].

Nonetheless, hypertension has a much higher prevalence compared to diabetes in reports from USA, Italy and China [5, 20]. In two meta-analyses from the same work group [24, 25], hypertension and diabetes were associated with poor outcomes in COVID-19. The association with hypertension was influenced by gender, but not by age or diabetes, while the association with diabetes was influenced by age and hypertension. On the other hand, in a study using a UK database [16], hypertension was associated with an increased risk of COVID-19 (HR 1.09) but there was no association when the risk was adjusted to age, gender, diabetes and obesity. In addition, diabetes and obesity were associated with an increased risk of death (HR 1.90 and 1.92, respectively).

Epidemiological data of COVID-19 in Mexico, indicate that obesity is a risk factor of death and is a partial mediator on the effects of diabetes in decreased survival [8]. Finally, when comorbidities in our sample were clustered, we observed a lower frequency of individuals with only-hypertension compared with only-diabetes and only-obesity patients. Therefore, diabetes and obesity should not be overlooked as strong risk factors and poor outcomes in individuals with COVID-19.

In our sample, the most frequent pharmacological treatment was hydroxychloroquine associated with azithromycin. Although there is evidence of an *in vitro* activity against SARS-CoV-2, the use of hydroxychloroquine and azithromycin for clinical benefit is only supported by limited and conflicting clinical data [26]. Antiviral agents were the second most frequently used treatment; for instance, oseltamivir was the third most administrated treatment; however, there are no reports about oseltamivir having in vitro activity against SARS-CoV-2. Although some clinical trials have included oseltamivir, it is not proposed as therapeutic intervention [27]. Lopinavir-ritonavir was less frequently used as treatment in our sample; while in vitro activity against SARS-CoV-2 has been reported, clinical trials have failed in associate it with better clinical outcomes [28]. Finally, the only immunomodulatory drug used in our sample was tocilizumab. In a meta-analysis, tocilizumab was associated with a significant decrease in mortality rate and ameliorate clinical symptoms in patients with COVID-19 [29]. Despite of their promising results, immunomodulatory drugs access is limited and they are not broadly used in clinical settings in Mexico.

Our study has some limitations that need to be considered. First, we had limited information about some individuals because they died within minutes after their arrival to the emergency room. Second, we did not include surviving individuals so we could not compare characteristics between groups. Third, we included data of only one frontline hospital for treating COVID-19 that only show characteristics and handling of patients at the beginning of the

COVID-19 pandemic in south Mexico; therefore, our data should not be applied to the whole Mexican population.

In conclusion, our study showed a high use of off-label of drugs that have insufficient evidence of their efficacy against COVID-19. The individuals in our sample had shorter hospital stays, higher frequency of shortness of breath, and higher prevalence of diabetes than similar populations from other countries. It is important to perform more clinical series and cohorts in Mexican population, as this population shows features that could differ of populations from other countries.

## Author Contributions

**Conceptualization:** Jesús Arturo Ruíz-Quiñonez.

**Data curation:** Crystell Guadalupe Guzmán-Priego, Guadalupe del Carmen Baeza-Flores.

**Formal analysis:** Carlos Alfonso Tovilla-Zarate.

**Investigation:** Oscar Israel Flores-Barrientos, Agustín Pérez-García.

**Methodology:** Carlos Ramón López-Brito.

**Project administration:** Jesús Arturo Ruíz-Quiñonez, Oscar Israel Flores-Barrientos, Víctor Narváez-Osorio.

**Resources:** Oscar Israel Flores-Barrientos, Víctor Narváez-Osorio.

**Software:** Carlos Alfonso Tovilla-Zarate.

**Supervision:** Crystell Guadalupe Guzmán-Priego, Carlos Ramón López-Brito, Carlos Alberto Denis-García, Agustín Pérez-García.

**Validation:** Carlos Alberto Denis-García.

**Visualization:** Guadalupe del Carmen Baeza-Flores, Carlos Ramón López-Brito.

**Writing – original draft:** Germán Alberto Nolasco-Rosales, Thelma Beatriz Gonzalez-Castro.

**Writing – review & editing:** Germán Alberto Nolasco-Rosales, Carlos Alfonso Tovilla-Zarate, Isela Esther Juárez-Rojop.

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
