## [Decision Letter · Decision Letter 0]

9 Nov 2020

PONE-D-20-29232

Features of patients who died of COVID-19 in a Hospital in the south of Mexico: A observational cohort study.

PLOS ONE

Dear Dr. Isela E Juárez-Rojop,

Thank you for submitting your manuscript to PLOS ONE. After careful consideration, we feel that it has merit but does not fully meet PLOS ONE’s publication criteria as it currently stands. Therefore, we invite you to submit a revised version of the manuscript that addresses the points raised during the review process.

We look forward to receiving your revised manuscript.

Kind regards,

Francesco Di Gennaro

Academic Editor

PLOS ONE

Journal Requirements:

2. Please provide additional details regarding participant consent. In the ethics statement in the Methods and online submission information, please ensure that you have specified (i) whether consent was informed and (ii) what type you obtained (for instance, written or verbal, and if verbal, how it was documented and witnessed).

If your study included minors, state whether you obtained consent from parents or guardians.

If the need for consent was waived by the ethics committee, please include this information.

3. Please ensure that you include a title page within your main document. You should list all authors and all affiliations as per our author instructions and clearly indicate the corresponding author.

Additional Editor Comments:

Dear Authors follow reviewer suggestion to improve your article

Reviewers' comments:

Reviewer's Responses to Questions

**Comments to the Author**

1. Is the manuscript technically sound, and do the data support the conclusions?

Reviewer #1: Partly

Reviewer #2: Yes

Reviewer #3: Yes

2. Has the statistical analysis been performed appropriately and rigorously? 

Reviewer #1: No

Reviewer #2: Yes

Reviewer #3: Yes

3. Have the authors made all data underlying the findings in their manuscript fully available?

Reviewer #1: No

Reviewer #2: Yes

Reviewer #3: Yes

4. Is the manuscript presented in an intelligible fashion and written in standard English?

Reviewer #1: No

Reviewer #2: Yes

Reviewer #3: Yes

5. Review Comments to the Author

Reviewer #1: I thank the authors for the opporunity to read their interesting work presenting findings of deceased COVID-19 patients.

However, I have some concerns regarding this study

1. English is poor please have a native speaker read through your text

2. COVID-19 as per death certificate does not necessarily represent cause of death or associated secondary complications that would be of interest to the reader. Similarly information on comorbidies is poor. How many of these patients were immunosuppressed ? What drugs were they receiving ? etc

3. Discussion of your findings in line with different epidemiological studies in other parts of the world is necessary. At the moment is just a report of your results

Reviewer #2: The manuscript is technical sound and statistical analysis performed appropriately. I feel the results should have also been expressed in a table format. The paper meets the PLOS ONE criteria for publication.

Reviewer #3: The manuscript is good even in Mexico there is not enough data publisheb on PubMed with by mexicans authors.

We know the lack of the real numbers of people afected by covid-19.

I think, this information could be different if we compare with the North of Mexico or even with Mexico City.

6. PLOS authors have the option to publish the peer review history of their article (what does this mean?). If published, this will include your full peer review and any attached files.

Reviewer #1: No

Reviewer #2: **Yes: **Victor Draman Afayo

Reviewer #3: No

---

## [Author Response · Author response to Decision Letter 0]

3 Dec 2020

Reviewer #1: I thank the authors for the opportunity to read their interesting work presenting findings of deceased COVID-19 patients. However, I have some concerns regarding this study

1. English is poor please have a native speaker read through your text

Done

2. COVID-19 as per death certificate does not necessarily represent cause of death or associated secondary complications that would be of interest to the reader. Similarly, information on comorbidies is poor. How many of these patients were immunosuppressed? What drugs were they receiving? Etc

We modified this information (Page 6, line 110-111; page 8, line 145-150) 

Change in the manuscript:

“All patients included in this study died of cardiorespiratory arrest, secondary to acute respiratory failure. […] Other less frequent comorbidities observed in individuals who died by COVID-19 are depicted in Table 3. There was one individual with HIV and other two with cancer; the three of them were immunosuppressed. It is important to mention that these individuals were also receiving the usual treatment for their underlying disease while they were treated in hospital for COVID-19.

3. Discussion of your findings in line with different epidemiological studies in other parts of the world is necessary. At the moment is just a report of your results.

We include information in Discussion section (Page 11; page 12, line 213, 214, 218-222).

Change in the manuscript:

Fever and cough had been reported as the most prevalent symptoms in case series from Europe, China, Mexico and in severe cases. On the other hand, the third most prevalent symptom varied widely; for instance, a review found dyspnea in 53 – 80% of cases, while a series of fatal cases in Wuhan found dyspnea in 70.6%. Nonetheless, in clinical series and meta-analysis involving non-lethal cases in Asia, Europe, and North America, the third most prevalent symptom was myalgias or fatigue (31-67%). […] In a follow-up study of 20,133 individuals with COVID-19 in United Kingdom, the most frequent comorbidities were chronic cardiac disease (31%), uncomplicated diabetes (21%), non-asthmatic chronic pulmonary disease (18%), and chronic kidney disease (16%); while only 23% reported no major comorbidities. On the other hand, a report in Northeast Brazil showed that the mortality of COVID-19 was enhanced by old age, neurological diseases, pneumopathies and cardiovascular diseases. […] Nonetheless, hypertension has a much higher prevalence compared to diabetes in reports from USA, Italy and China. […] On the other hand, in a study using a UK database, hypertension was associated with an increased risk of COVID-19 (HR 1.09) but there was no association when the risk was adjusted to age, gender, diabetes and obesity. In addition, diabetes and obesity were associated with an increased risk of death (HR 1.90 and 1.92, respectively).

Reviewer #2: The manuscript is technical sound and statistical analysis performed appropriately. I feel the results should have also been expressed in a table format. The paper meets the PLOS ONE criteria for publication.

We modified this point, we included Table 2, 3 and 4 (page 7, line 132; page 9, line 156 and 164).

Reviewer #3: The manuscript is good even in Mexico there is not enough data published on PubMed with by mexicans authors. We know the lack of the real numbers of people affected by covid-19. I think, this information could be different if we compare with the North of Mexico or even with Mexico City.

Response 6. We included evidence about Mexico data (page 10, line 180-182; page 11, line 203-206)

Change in the manuscript:

Similar findings had been observed in the state of Sonora, Northwest of Mexico, with a mean age of 56.98 years old in individuals who died by COVID-19. […] It has been reported that in Mexican individuals with COVID-19 there is a slightly higher frequency of hypertension than diabetes. Additionally, a report from Northwest Mexico indicated that diabetes and hypertension have higher risk of mortality than other comorbidities.

6. PLOS authors have the option to publish the peer review history of their article (what does this mean?). If published, this will include your full peer review and any attached files.

Do you want your identity to be public for this peer review? For information about this choice, including consent withdrawal, please see our Privacy Policy.

Reviewer #1: No

Reviewer #2: Yes: Victor Draman Afayo

Reviewer #3: No

---

## [Decision Letter · Decision Letter 1]

30 Dec 2020

Features of patients that died for COVID-19 in a Hospital in the south of Mexico: A observational cohort study.

PONE-D-20-29232R1

Dear Dr. Isela,

We’re pleased to inform you that your manuscript has been judged scientifically suitable for publication and will be formally accepted for publication once it meets all outstanding technical requirements.

Kind regards,

Francesco Di Gennaro

Academic Editor

PLOS ONE

Additional Editor Comments (optional):

dear authors congratulations

Reviewers' comments:

Reviewer's Responses to Questions

**Comments to the Author**

1. If the authors have adequately addressed your comments raised in a previous round of review and you feel that this manuscript is now acceptable for publication, you may indicate that here to bypass the “Comments to the Author” section, enter your conflict of interest statement in the “Confidential to Editor” section, and submit your "Accept" recommendation.

Reviewer #2: All comments have been addressed

2. Is the manuscript technically sound, and do the data support the conclusions?

Reviewer #2: Yes

3. Has the statistical analysis been performed appropriately and rigorously? 

Reviewer #2: Yes

4. Have the authors made all data underlying the findings in their manuscript fully available?

Reviewer #2: Yes

5. Is the manuscript presented in an intelligible fashion and written in standard English?

Reviewer #2: Yes

6. Review Comments to the Author

Reviewer #2: (No Response)

7. PLOS authors have the option to publish the peer review history of their article (what does this mean?). If published, this will include your full peer review and any attached files.

Reviewer #2: No

---

## [Editor Report · Acceptance letter]

1 Feb 2021

PONE-D-20-29232R1 

Features of patients that died for COVID-19 in a Hospital in the south of Mexico: A observational cohort study. 

Dear Dr. Juárez-Rojop:

I'm pleased to inform you that your manuscript has been deemed suitable for publication in PLOS ONE. Congratulations! Your manuscript is now with our production department. 

Kind regards, 

on behalf of

Dr. Francesco Di Gennaro 

Academic Editor

PLOS ONE